# Epidemiological, socio-demographic and clinical features of the early phase of the COVID-19 epidemic in Ecuador

**Esteban Ortiz-Prado**[1]*, **Katherine Simbaña-Rivera**[1], **Lenin Gómez Barreno**[1], **Ana Maria Diaz**[1], **Alejandra Barreto**[1], **Carla Moyano**[1], **Vannesa Arcos**[1], **Eduardo Vásconez-González**[1], **Clara Paz**[2], **Fernanda Simbaña-Guaycha**[3], **Martin Molestina-Luzuriaga**[1], **Raúl Fernández-Naranjo**[1], **Javier Feijoo**[4], **Aquiles R. Henriquez-Trujillo**[1], **Lila Adana**[2], **Andrés López-Cortés**[5,6], **Isabel Fletcher**[7,8], **Rachel Lowe**[7,8,9]

**1** One Health Research Group, Faculty of Health Science, Universidad de Las Americas, Quito, Ecuador, **2** School of psychology, Universidad de Las Americas, Quito, Ecuador, **3** Scientific Association of Medical Students, Universidad Central del Ecuador, Quito, Ecuador, **4** Instituto de Física La Plata, Universidad Nacional de la Plata, La Plata, Argentina, **5** Centro de Investigación Genética y Genómica, Facultad de Ciencias de la Salud Eugenio Espejo, Universidad UTE, Quito, Ecuador, **6** Red Latinoamericana de Implementación y Validación de Guías Clínicas Farmacogenómicas (RELIVAF-CYTED), Quito, Ecuador, **7** Centre for Mathematical Modelling of Infectious Diseases, London School of Hygiene & Tropical Medicine, London, United Kingdom, **8** Centre on Climate Change and Planetary Health, London School of Hygiene & Tropical Medicine, London, United Kingdom, **9** Barcelona Institute for Global Health (ISGlobal), Barcelona, Spain

* e.ortizprado@gmail.com

**Data Availability Statement:** The data underlying the results presented in the study are available from the MoH of Ecuador under written request the

## Abstract

The SARS-CoV-2 virus has spread rapidly around the globe. Nevertheless, there is limited information describing the characteristics and outcomes of COVID-19 patients in Latin America. We conducted a cross-sectional analysis of 9,468 confirmed COVID-19 cases reported in Ecuador. We calculated overall incidence, mortality, case fatality rates, disability adjusted life years, attack and crude mortality rates, as well as relative risk and relative odds of death, adjusted for age, sex and presence of comorbidities. A total of 9,468 positive COVID-19 cases and 474 deaths were included in the analysis. Men accounted for 55.4% (n = 5, 247) of cases and women for 44.6% (n = 4, 221). We found the presence of comorbidities, being male and older than 65 years were important determinants of mortality. Coastal regions were most affected by COVID-19, with higher mortality rates than the highlands. Fatigue was reported in 53.2% of the patients, followed by headache (43%), dry cough (41.7%), ageusia (37.1%) and anosmia (36.1%). We present an analysis of the burden of COVID-19 in Ecuador. Our findings show that men are at higher risk of dying from COVID-19 than women, and risk increases with age and the presence of comorbidities. We also found that blue-collar workers and the unemployed are at greater risk of dying. These early observations offer clinical insights for the medical community to help improve patient care and for public health officials to strengthen Ecuador's response to the outbreak.

data will be provided to anyone requesting it. The information can be obtained upon formal request to the following e-mail address: boletin. dis@mspsalud.gob.ec.

**Funding:** Funding was received from Universidad de las Americas. RL was supported by a Royal Society Dorothy Hodgkin Fellowship. The funders had no role in study design, data collection and analysis, decision to publish, or preparation of the manuscript.

**Competing interests:** The authors have declared that no competing interests exist.

## Author summary

In this study we summarize the epidemiological trends of the early phase of the novel coronavirus disease (COVID-19) pandemic in Ecuador, the country with the highest excess mortality reported at the beginning of the global health crisis due to COVID-19 worldwide.

We have carried out a complete analysis of the epidemiological trends, clinical features, risk factors associated with death and the main demographic characteristics of the first 9,468 patients. Ecuador has officially reported 474 COVID-19 confirmed deaths, nevertheless, at least 4,780 deaths were reported as acute respiratory distress syndrome (ARDS) during the same period of time, suggesting an important underreporting and undertesting of COVID-19 cases in the country. In Ecuador, COVID-19 is five times more lethal among unemployed patients compared to white-collar workers, suggesting a strong association between poverty and mortality. This is the first epidemiological report from Ecuador that uses the official reported cases of COVID-19. High altitude dwellers had lower attack and mortality rates than those living in lowlands, possibly due to reduced access to health care services.

## Introduction

For the past few decades, the world has been exposed to a series of threats from viral outbreaks caused by emerging zoonotic diseases and in particular by a family of viruses known as coronaviruses [1]. The World Health Organization recognizes at least three types of coronavirus capable of generating epidemic outbreaks, including SARS-CoV, MERS-CoV and the recently discovered SARS-CoV-2 virus [2]. These viruses are responsible for causing severe acute respiratory syndrome (SARS), Middle East respiratory syndrome (MERS) and the most recently described coronavirus disease (COVID-19) [3].

Since the first reports of a cluster of atypical pneumonia cases in Wuhan, China on December 2019, the SARS-CoV-2 virus and COVID-19 has quickly spread across the globe [1,4], infecting more than 40 million people and causing more than 1.1 million **deaths** worldwide until October 20[th], 2020. One of the main reasons the virus has spread so rapidly is due to droplet transmission from both symptomatic and asymptomatic people, making it difficult to test, trace and isolate new cases effectively [2].

Understanding the transmission dynamics in different settings can provide important clues about the advance of the pandemic, especially in areas with unequal access to health services, high population density and a high burden of neglected tropical diseases [5,6].

In this study, we present the findings of an interim analysis of the epidemiological situation in Ecuador, describing the clinical characteristics and epidemiological behavior of the first 9,468 confirmed COVID-19 cases using the reverse transcription polymerase chain reaction (RT-PCR) method.

## Methods

### Ethics statement

This secondary data analysis of anonymized, un-identifiable information received ethical approval (**#EOP-200301-001**) from the Universidad de las Americas Ethics Committee CEISH on 10[th] of March 2020. The phone calls and the information retrieved during the follow-up period was obtained by health professionals not linked to this project as part of their

standard of care treatment for patients with COVID-19. The information from those reports were posteriorly anonymized ensuring no sensitive information was shared with any member of our research project. According to the international good clinical and research practices and in accordance with Ecuadorian law, observational studies that do not jeopardize the rights of patients are exempt from obtaining full ethical approval.

## Study design

We conducted a country-wide population-based analysis of the epidemiology of COVID-19, using case data reported in Ecuador between 27 February and 18 April 2020 (S1 Fig). The database comprised men, women and children from 0 to 100 years old with a positive RT-PCR COVID-19 diagnosis, using a reverse transcription polymerase chain reaction (RT-PCR) technique, during the first 54 days of the outbreak, including the first imported and the subsequent community-acquired cases.

## Setting

The study was conducted in Ecuador, a country located in South America, bordering with Colombia to the North, Peru to the South/East and the Pacific Ocean to the West. The country is divided into four geographical regions: 1) the coastal region, 2) the highlands or sierra region, 3) the Amazon region and 4) the insular region (Galapagos Islands). The population of Ecuador was estimated to be 17,510,643 inhabitants based on the latest available projections for 2020[7]. Ethnicity in Ecuador is distributed as follows: 7.1% indigenous (1,059,863), 7.2% Afro Ecuadorian (1,077,878), 79.3% Mestizo (11,910,702), 6.1% self-reported white or Caucasians (909,741) and 0.4% other groups according to the 2010 national census [8].

## Data

We obtained socio-demographic variables, such as age, sex, marital status and place of residence from the Ministry of Health (MoH) registries. Clinical data including date of onset of symptoms, date of diagnosis and date of death, as well as the presence of comorbidities, pregnancy and influenza vaccination history were also obtained. Epidemiological information including city and province of registration, elevation, occupation, travel history and institution of diagnosis were analyzed. We also obtained the results of tests performed nationwide up until 18th April 2020 and developed testing trends per days, including positive, negative, suspected and not processed tests. The information was transferred to our research team after presenting a formal petition and signing a confidentiality agreement with the MoH to protect patient's rights.

From the online self-reporting tool delivered through the MoH surveillance department, clinical information about signs and symptoms, as well as civil status and educational attainment were obtained from the 856 outpatients (COVID-19 positive) that completed the voluntary self-reporting tool.

The number of deaths used in this analysis were officially reported as COVID-19 using the 10th revision of the International Statistical Classification of Diseases and Related Health Problems (ICD) code: U07.1 COVID-19, virus identified, U07.2 COVID-19, virus not identified.

In order to estimate the number of additional deaths that were not diagnosed as COVID-19 during March and April 2020, we compared the COVID-19 database with the National Death Registry, including fatalities registered as Acute Respiratory Distress Syndrome (ICD J80).

## Study size and sample size calculation

This study included 9,468 RT-PCR confirmed COVID-19 patients and the first 474 officially reported deaths.

In terms of the number of patients required ($x$) to complete the self-reported symptoms tool distributed by the MoH, the sample size ($n$) and margin of error ($E$) were given by the following formula:

$$x = Z(^c/_{100})^2 r(100-r)$$

$$n = {}^{Nx}/_{((N-1)E^2 + x)}$$

$$E = Sqrt[^{(N-n)x}/_{n(N-1)}]$$

Where $N$ is the population size (9,468), ($r$) is the fraction of expected responses (50%), and $Z(c/100)$ is the critical value for the confidence level ($c$).

The total number of completed answers required according to the calculations was 621 in order to achieve a 99% confidence level. Through a convenience-based sampling technique 856 patients were included. The self-reporting tool was sent to all patients not hospitalized who were under epidemiological surveillance by the MoH and we included only those who voluntarily agree to provide information on educational attainment, symptomatology and civil status.

## Descriptive statistics

The overall incidence (attack rate), mortality rate, and case fatality rate were computed according to the entire population at risk living in a canton or a province. Information from the first 9,468 COVID-19 confirmed patients in Ecuador were compared with the population at risk, obtained from the publicly available canton- and province-level sex-specific projections. Measurements of frequency (counts, absolute and relative percentages), central tendency (median), dispersion (interquartile range) and absolute differences were calculated for all categorical and continuous variables. Case fatality rates (CFR%) were computed using the number of cases officially reported while the dichotomized variable (alive or died) was used as an outcome of the confirmed cases. CFR% were adjusted to mitigate the effect in crude CFR% estimation. We adjusted the naive estimates of CFR% to account for the delay from hospitalization-to-death for those cases that were fatal using a delay-adjusted case fatality ratio proposed by Russel et.al 2020.

Positive test rates (PTR%) were calculated using the number of positive test results divided by the total number of tests conducted. The attack and crude mortality rates per canton and province were computed per 100,000 inhabitants. Important landmark dates are included in S5 Fig.

## Statistical analysis

To examine associations, relative risks (RR) and 95% confidence intervals were computed for all age groups using female cases as the reference level. We performed a relative risk (RR) analysis of the total number of expected cases by the population at risk in all the groups to obtain the likelihood of dying due to COVID-19. In order to control for the effects of sex, age and comorbidities, an adjusted logistic regression was performed using the final outcome (alive or died) as a response variable. The geographical distribution of residence was divided into high

($>$ = 2,500 m) and low altitude ($<$ 2,500 m), in order to analyze the differences within these populations using a t-test with a 0.05 significance level.

### Burden of disease

The number of years of healthy life lost due to COVID-19 among the reported cases were calculated using Disability-Adjusted Life Years (DALY). DALYs are the sum of the Years of Life Lost (YLL) due to premature mortality in the population and the Years Lost due to Disability (YLD) caused by the consequences of the disease [9]. YLL were calculated from the number of deaths per age group multiplied by the standard life expectancy at the age of death. We used the life expectancy table developed by the 2010 Global Burden of Disease study, with a life expectancy at birth of 86 for both males and females [10]. To estimate YLD, the total number of cases in the study period was multiplied by the average duration of the disease reported by the International Severe Acute Respiratory and Emerging Infections Consortium (ISARIC) COVID-19 database and the disability weights corresponding to the different degrees of severity of infectious diseases included in the 2013 Global Burden of Disease study [11,12].

## Results

There were 9,468 positive cases of COVID-19 and 474 officially reported deaths in Ecuador from 27 February– 18 April 2020 (54-day period). 99.3% of COVID-19 patients were Ecuadorians (n = 9,400), 0.30% (n = 44) were from other countries in Latin America and the other 0.40% (n = 24) were either from Europe, North America or Asia.

### Age and sex analysis

Men accounted for 55.4% (n = 5,247) of all cases with an incidence rate of 60.5 per 100,000, while women accounted for 44.6% of cases (n = 4,221) and an incidence of 47.2 per 100,000. The median age of COVID-19 patients was 42 (IQR: 32–56) in men and 39 (IQR: 30–54) in women. The median age of patients who had died from COVID-19 was 62 (IQR: 51–70) in men and 65 (IQR: 56–74) in women.

We also found that men were more likely to die from COVID-19 in almost every age group (Fig 1), although case fatality rate for patients aged between 85–94 was higher amongst women (Table 1).

### Socio-demographic variables

**Ethnicity.** The majority of COVID-19 cases occurred among Mestizos 78% (n = 7,367), followed by indigenous 0.79% (n = 75), Caucasians 0.84% (n = 40) and Afro-Ecuadorians/ Black with ~0.1% (n = 16) (Table 2).

**Education attainment and marital status.** In Ecuador, the results of the demographic survey (n = 856) show that 0.7% (n = 6) of COVID-19 patients had not completed their elementary school, 1.9% (n = 16) reported a complete elementary school education, 5.8% (n = 50) had not completed their high school education and 22.5% (n = 193) had completed their entire high school cycle. At least 6.8% (n = 58) are technicians, 46.3% (n = 396) have completed their undergraduate degree and 16.0% (n = 137) have formal postgraduate training. In terms of civil status 39.3% (n = 336) are single, 38% (n = 327) are married, 13% (n = 114) are cohabitating with their partners, 5.4% are divorced (n = 46), 1.3% widows (n = 11) and other responses 3% (n = 22).

**Occupation and work-related risk.** The results of COVID-19 mortality by occupation showed that 19% of infected people (n = 1,800) are health professionals (Table 3). From this

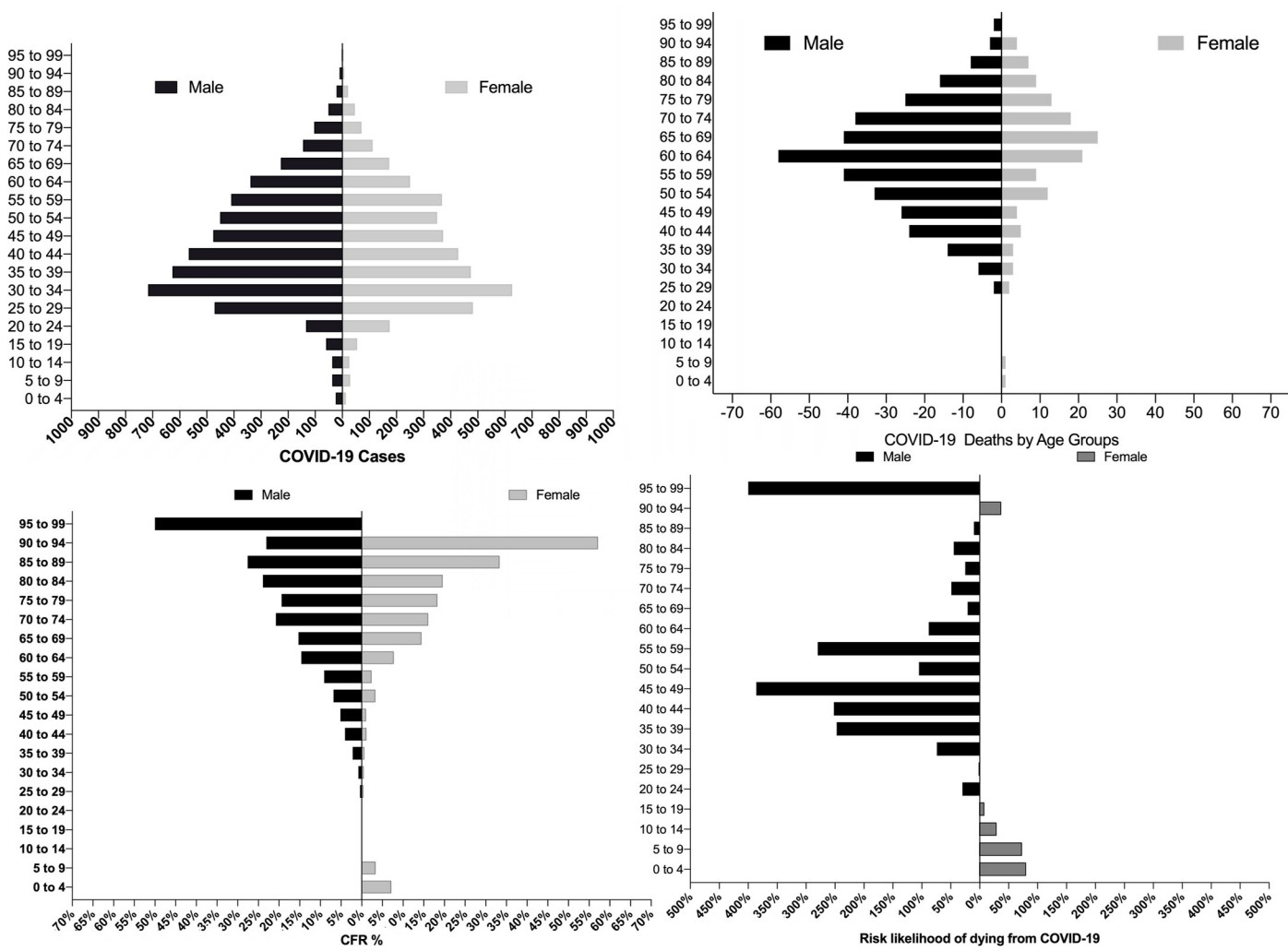

**Fig 1. COVID-19 cases and deaths in Ecuador by age and sex.** Top panel: Age pyramids for confirmed cases and officially reported deaths. Lower panel: Case fatality rate (%) and risk likelihood of dying from COVID-19, for females (blue) and males (purple).

group medical doctors (n = 876) were the most affected professionals in Ecuador, representing at least 9.3% of all reported cases (Fig 2)

Nurses are the second most affected group with 3.3% representing at least 309 people (Table 2). The majority of people testing positive work in manual labor (n = 1,390). In terms of fatality, the small number of inmates seem to be at the highest risk of dying, with a RR of 7.75 [p value: 0.020, 1.3–44.0], followed by retired elderly people with a RR of 5.6 [p value: <0.0001, 3.5–9.0] and unemployed with 4.47 [p value: <0.0001, 2.8–6.9]. The results from the likelihood of dying is increasingly higher among 'blue collar' workers, civil servants (police officers, soldiers, members of the navy, firefighters and other agents), unemployed, retirees and prisoners (Table 3). Although a small number of cases were reported among politicians (n = 18), the CFR% for this group was zero, followed by nurses (0.3%), other health care professionals (0.7%) and medical doctors with a CFR% of 1.6%.

**Vaccination history, comorbidities and pregnancy history among COVID-19 patients.** The current analysis of the immunization history of influenza showed that 99.1% of patients (n = 9,384) did not report any vaccination history in the last year and only 0.9%

**Table 1. Case fatality rate (CFR%) due to COVID-19 and relative risk (RR) in Ecuador among women and men from different age groups (N = 9,468).** *P-values below or equal to 0.05 were considered significant.

| | Female | | | Male | | | | | | | |
| Age | Alive | Death | CFR% | Alive | Death | CFR% | RR | Lower limit (CI 95%) | Upper limit (CI 95%) | P Value | Risk |
|---|---|---|---|---|---|---|---|---|---|---|---|
| 0 to 4 | 13 | 1 | 7.14% | 24 | 0 | 0.00% | 0.2 | 0.0087 | 4.6023 | 0.3145 | -80% |
| 5 to 9 | 29 | 1 | 3.33% | 37 | 0 | 0.00% | 0.27 | 0.0115 | 6.4437 | 0.806 | -73% |
| 10 to 14 | 26 | 0 | 0.00% | 37 | 0 | 0.00% | 0.71 | 0.0145 | 4.7119 | 0.8632 | -29% |
| 15 to 19 | 55 | 0 | 0.00% | 60 | 0 | 0.00% | 0.92 | 0.0185 | 5.4976 | 0.9657 | -8% |
| 20 to 24 | 175 | 0 | 0.00% | 134 | 0 | 0.00% | 1.3 | 0.026 | 9.2868 | 0.8943 | 30% |
| 25 to 29 | 482 | 2 | 0.41% | 471 | 2 | 0.42% | 1.02 | 0.1447 | 7.2347 | 0.9816 | 2% |
| 30 to 34 | 627 | 3 | 0.48% | 717 | 6 | 0.83% | 1.7427 | 0.4376 | 6.9397 | 0.4308 | 74% |
| 35 to 39 | **474** | **3** | **0.63%** | **627** | **14** | **2.18%** | **3.47** | **1.0036** | **8.0162** | **0.0493*** | **247%** |
| 40 to 44 | **428** | **5** | **1.15%** | **567** | **24** | **4.06%** | **3.516** | **1.3526** | **9.1436** | **0.0099*** | **252%** |
| 45 to 49 | **372** | **4** | **1.06%** | **476** | **26** | **5.18%** | **4.86** | **1.7136** | **11.8316** | **0.003*** | **386%** |
| 50 to 54 | **350** | **12** | **3.31%** | **451** | **33** | **6.82%** | **2.05** | **1.0775** | **3.9263** | **0.0288*** | **105%** |
| 55 to 59 | **368** | **9** | **2.39%** | **410** | **41** | **9.09%** | **3.80** | **1.8752** | **7.7332** | **0.0002*** | **280%** |
| 60 to 64 | **250** | **21** | **7.75%** | **339** | **58** | **14.61%** | **1.88** | **1.1728** | **3.0307** | **0.0088*** | **88%** |
| 65 to 69 | 173 | 25 | 14.45% | 227 | 41 | 15.30% | 1.21 | 0.7632 | 1.9235 | 0.4156 | 21% |
| 70 to 74 | 112 | 18 | 16.07% | 145 | 38 | 20.77% | 1.49 | 0.8972 | 2.5069 | 0.1221 | 49% |
| 75 to 79 | 71 | 13 | 18.31% | 104 | 25 | 19.38% | 1.25 | 0.6795 | 2.3076 | 0.4708 | 25% |
| 80 to 84 | 46 | 9 | 19.57% | 51 | 16 | 23.88% | 1.45 | 0.7 | 3.0425 | 0.3132 | 45% |
| 85 to 89 | 21 | 7 | 33.33% | 21 | 8 | 27.59% | 1.10 | 0.4616 | 2.6375 | 0.8248 | 10% |
| 90 to 94 | 7 | 4 | 57.14% | 10 | 3 | 23.08% | 0.6346 | 0.1794 | 2.2449 | 0.4805 | -37% |
| 95 to 99 | 5 | 0 | 0.00% | 2 | 2 | 50.00% | 6.00 | 0.3667 | 11.1622 | 0.2089 | 500% |
| Total | 4,084 | 137 | 3.35% | 4,910 | 337 | 6.42% | 1.9789 | 1.6292 | 2.4035 | 0.0001 | 98% |

(n = 84) of patients reported having been vaccinated. From the not vaccinated group, 471 patients died while from the vaccinated group, only 3 died. The risk of dying among the not vaccinated group was higher than the vaccinated group, although the difference was not statistically significant (1.40 [95% CI: 0.46 to 4.28]) (Table 2).

In our study, 1.4% of women (n = 60) were pregnant and 4 of them died (6.7%) due to COVID-19. The RR of pregnant women dying was not found to be statistically significant (RR: 2.05 [0.785 to 5.36] p value: 0.14).

The median age of COVID-19 patients reporting comorbidities in Ecuador was 59 (IQR: 49–68) years old,while those with no comorbidities had a median age of 42 years old (IQR: 32–55). When the presence of comorbidities was incorporated in an adjusted logistic regression model, the presence of comorbidities, being male and older than 65 years old increased the risk of dying by almost 130% (OR: 2.27 [1.72–3.00, p value <0.001]), 100% (OR: 2.03[1.65–2.50, p value < 0.001]) and 470% (OR: 5.74 [4.7–7.0, p value <0.001]) (Table 4).

**Symptoms assessment.** The most common symptom reported among COVID-19 patients in Ecuador was fatigue or general tiredness (53.2%), followed by headaches (43%), and dry cough (41.7%). 37.1% of the patients reported loss of taste (ageusia), 36.1% reported loss of smell (anosmia) and 35% reported muscle and joint pain (S2 Fig). The median time elapsed between the onset of symptoms and the time before receiving medical attention was four days (IQR: 1–8 days). The average time between onset of symptoms in patients and case notification was nine days (IQR: 5–14 days). When observed retrospectively using the date of the onset of symptoms, 29 undocumented patients were already sick but only 6 were diagnosed, a trend that continued to be high until March 24th, the day when strict government interventions were implemented in Ecuador (Fig 3).

**Table 2. Total number of deaths, patients alive and percentage of total (%) and case fatality rate (%) for women and men in different ethnic groups, type of health-care provision, presence of comorbidities and travel history.**

| | Women | | | Men | | |
|---|---|---|---|---|---|---|
| | Deaths (n = 137) | Alive (n = 4,084) | CFR (%) | Deaths (n = 337) | Alive (n = 4,910) | CFR (%) |
| **Ethnicity (% of population)** | | | | | | |
| Afro Ecuadorian (7.2%) | 0 (0%) | 4 (0.1%) | 0% | 0 (0%) | 2 (0%) | 0% |
| Caucasian (6.1%) | 1 (0.7%) | 16 (0.4%) | 5.9% | 0 (0%) | 23 (0.5%) | 0% |
| Indigenous (7%) | 2 (1.5%) | 30 (0.7%) | 6.3% | 5 (1.5%) | 38 (0.8%) | 11.6% |
| Mestizo/a (71.9%) | 128 (93.4%) | 3,197 (78.3%) | 3.8% | 314 (93.2%) | 3,728 (75.9%) | 7.8% |
| Montubio (7.4%) | 3 (2.2%) | 39 (1%) | 7.1% | 10 (3%) | 40 (0.8%) | 20% |
| Mulato | 0 | 3 (0.1%) | 0.0% | 0 (0%) | 2 (0%) | 0% |
| Black | 0 | 4 (0.1%) | 0.0% | 0 (0%) | 6 (0.1%) | 0% |
| Others (0.3%) | 1 (0.7%) | 8 (0.2%) | 11.1% | 0 (0%) | 6 (0.1%) | 0% |
| No data | 2 (1.5%) | 783 (19.2%) | 0.3% | 8 (2.4%) | 1,065 (21.7%) | 0.7% |
| **Healthcare provider** | | | | | | |
| Private | 6 (4%) | 995 (24.4%) | 0.6% | 12 (3.6%) | 1,424 (29%) | 0.8% |
| Public | 131 (96%) | 3,089 (75.6%) | 4.1% | 325 (96.4%) | 3,486 (71%) | 8.5% |
| **Presence of comorbidities** | | | | | | |
| Yes | 22 (16.1%) | 191 (4.7%) | 10.3% | 55 (16.3%) | 270 (5.5%) | 16.9% |
| No | 115 (83.9%) | 3,893 (95.3%) | 2.9% | 282 (83.7%) | 4,640 (94.5%) | 5.7% |
| **Travel history** | | | | | | |
| Yes | 2 (1.5%) | 66 (1.6%) | 2.9% | 9 (2.7%) | 64 (1.3%) | 12.3% |
| No | 55 (40.1%) | 1160 (28.4%) | 4.5% | 139 (41.2%) | 1496 (30.4%) | 8.5% |
| No data | 80 (58.4%) | 2,858 (70.0%) | 2.7% | 189 (56.1%) | 3350 (68.2%) | 5.3% |

## Epidemiological analysis

**Contact tracing.** The results from the self-reporting tool (n = 856) suggested that 42.6% (n = 365) of patients did not have contact with anyone after diagnosis, 11.8% (n = 101) might have contacted 1 to 5 people, 4.9% (n = 42) might have contacted 6 to 10 people, and 5.8% (n = 50) might have contacted more than 10 people after diagnosis. The other 34.9% (n = 299) of the cohort did not recall such information.

**Travel history and nationality.** In Ecuador, 99.3% of the patients were Ecuadorians (n = 9,400), 0.30% (n = 44) were from Latin-America and the other 0.40% (n = 24) were either from Europe, North America or Asia. In terms of the number of imported cases, at least 11 patients with confirmed COVID-19 had travelled from 5 countries, while at least 132 cases had a recognized epidemiological connection with people who had travelled to at least one of more than 25 different countries (S3 Fig)

**Days since diagnosis.** Among 9,468 patients, the median time elapsed between the onset of symptoms and the time of medical attention was 4 days (IQR: 1–8 days). However, from the onset of symptoms to the day of notification 9 days elapsed (IQR: 5–14 days).

Only 6 cases had been diagnosed at the end of February. However, when observed retrospectively using the date of the onset of diagnosis, 29 patients were already sick (Fig 3). Unconfirmed cases continued to be high until 24th of March, the day when a lock-down was implemented in Ecuador (Fig 3).

The median time elapsed from the onset of symptoms to the day of death was 11 days (IQR: 7–15 days) for the 474 patients that died due to COVID-19 in Ecuador (S6 Fig). The median time between first day of medical attention until death was 5 days (IQR: 2–8 days). The median time between case notification and death was 4 days (IQR: 2–7 days).

**Table 3. Case fatality rate (%), relative risk and risk likelihood due to COVID-19 in Ecuador by occupation.** Relative risk was estimated using White-Collar jobs as a reference group. P-values below or equal to 0.05 were considered significant.

| Occupation | Deaths | Cases | CFR% | Relative Risk | p-value | Lower limit (CI 95%) | Upper limit (CI 95%) | Likelihood of outcome |
|---|---|---|---|---|---|---|---|---|
| **Politicians*** | 0 | 18 | 0.0% | 0.016 | **0.004** | **0.001** | **0.273** | -98% |
| **Registered Nurses** | 1 | 309 | 0.3% | 0.1 | **0.023** | **0.013** | **0.732** | -90% |
| **Health Care professionals no physicians**** | 4 | 583 | 0.7% | 0.211 | **0.003** | **0.074** | **0.600** | -79% |
| **Medical doctors** | 14 | 876 | 1.6% | 0.487 | **0.027** | **0.257** | **0.923** | -51% |
| **Dentists** | 1 | 32 | 3.1% | 0.939 | 0.062 | 0.131 | 6.705 | -6% |
| **White Collar Jobs***** | 27 | 810 | 3.3% | Ref. | Ref. | Ref. | Ref. | Ref. |
| **Civil Services**° | 16 | 324 | 4.9% | 1.458 | 0.221 | 0.796 | 2.672 | 46% |
| **Others** | 26 | 405 | 6.4% | 1.870 | **0.019** | **1.10** | **3.164** | 87% |
| **Blue Collar**† | 186 | 1,390 | 13.4% | 3.658 | **<0.001** | **2.465** | **5.433** | 266% |
| **Unemployed** | 53 | 314 | 16.9% | 4.476 | **<0.001** | **2.863** | **6.999** | 348% |
| **Retired** | 39 | 175 | 22.3% | 5.649 | **<0.001** | **3.541** | **9.013** | 465% |
| **Prisoners** | 1 | 3 | 33.3% | 7.75 | **0.020** | **1.363** | **44.04** | 675% |
| **No Information** | 106 | 4,229 | 2.5% | N/A | N/A | N/A | N/A | N/A |
| **Total** | 474 | 9,468 | 5.0% | N/A | N/A | N/A | N/A | N/A |

N/A = Not applied

*Diplomatic, city mayor, political advisor and city/provincial councilors

** Hospital and minister of public health administrative personnel, medical student nursing assistant, hospital cellars, medical assistant, therapists, laboratory, x-ray and pharmacy technician, diagnostic medical sonographer, primary health care technicians (TAPS), physiotherapist, obstetricians and paramedics.

***Engineers, lawyer, auditors, psychologist, chefs, architect, communicator, teachers, managers, chief executive officers, entrepreneurs, analysts, administrator.

°Military soldiers and army staff, navy personal, transit agents, policemen, security guards, jail security agents.

†Informal merchants, kitchen staff, housework, domestic services employees, informal driver, farmers, craftsman, machine operators, cleaning services employees, mechanics, carpenters and construction workers.

**Attack rate (confirmed cases per 100,000 people).** The overall attack rate was 51.1 per 100,000 people. Sex-specific attack rates of COVID-19 was 60.5 per 100,000 for men and 47.2 per 100,000 for women. In terms of age-adjusted attack rates per 100,000, the lowest attack rate was found among children from 0–4 years old (2.86 per 100,000), while the highest attack rate was found in patients aged between 55 and 59 years (111 per 100,000).

**Mortality rates (confirmed deaths per 100,000 people).** The overall confirmed mortality rate was 2.7 per 100,000 people. This value ranged from 0.1 to 20.1 per 100,000 people at risk, with 90–95 years of age being the most affected group with a crude age specific mortality rate of 20 per 100,000 people. From the official reports, 474 deaths were recorded as COVID-19 (RT-PCR confirmed).

During March and April 2020, Ecuador experienced a unique increment in its overall mortality. In 2018, at least 23,973 deaths were officially reported between January to April, in the following year, 25,061 and during January to April 15[th], 2020, 29,392 were registered, indicating an increase in more than 15% compared with the previous year. The increase over the same period from 2018 to 2019 was just 4.4%.

We analyzed the total number of deaths included in the National Registry. From March 1[st] to April 18[th], 2020, 4,780 reported deaths were related to acute respiratory distress syndrome (ARDS) in Ecuador (37%). From the total of ARDS-related deaths, 1,283 were probable COVID-19 (U07.2 clinically-epidemiologically diagnosed), 809 were registered as suspected COVID-19 (U07.2) and 474 confirmed COVID-19 cases (U07.1).

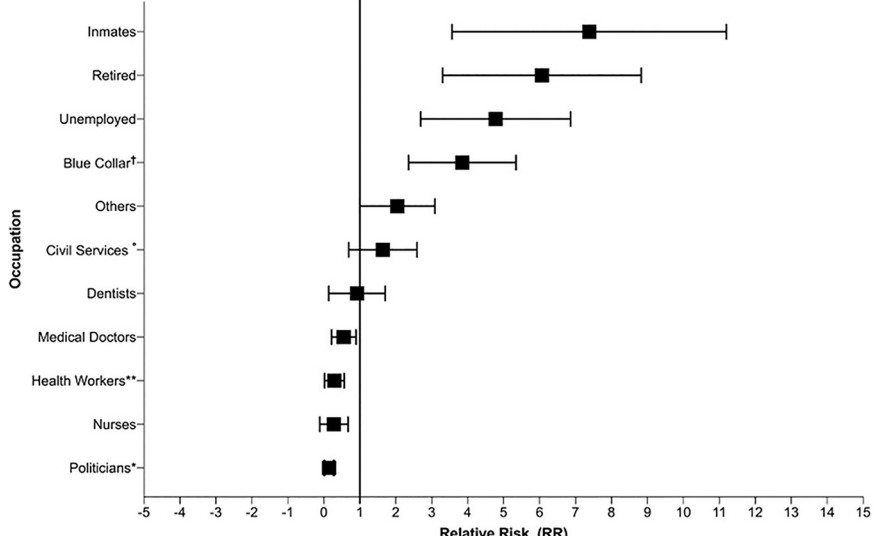

**Fig 2. Relative risk calculation by occupation compared with white-collar workers (Engineers, lawyer, auditors, psychologists, chefs, architects, communicator, teachers, managers, chief executive officers, entrepreneurs, analysts, administrator).** *Diplomatic, City Mayor, political advisor and city/provincial councilors, ** Hospital and Minister of Public Health administrative personnel, Medical Student nursing assistant, hospital cellars, Medical Assistant, Therapists, Laboratory, X-ray and Pharmacy Technician, Diagnostic Medical Sonographer, primary health care technicians (TAPS), physiotherapist, obstetricians and paramedics. *Military soldiers and army staff, Navy personal, Transit agents, Policemen, Security guards, Jail security agents. †Informal merchants, kitchen staff, Housework, Domestic Services employees, informal driver, Farmers, Craftsman, Machine operators, Cleaning Services employees, mechanics, carpenters and construction workers.

We found an 11-day lag (IQR: 7–15 days) between COVID-19 cases symptoms onset and the reported number of COVID-19 related deaths (S4 Fig).

**Basic reproduction number (R0).** We calculated the basic reproduction number (R0) for the entire country using the exponential growth criteria. During the first months of the pandemic while numbers were increasing exponentially, the R0 was found to be 3.54 (CI: 3.46–3.63). When computing the R0 by provinces, we found R0 values as high as 3.68 (CI: 3.18–4.28) in Los Rios, followed by 3.68 (CI: 2.53–5.28) in Esmeraldas and 3.67 (CI: 3.56–3.78) in

**Table 4. Adjusted logistic regression using final outcome (Dead/Alive) as a dependent variable and the presence of comorbidities, sex and age of the patients as a covariate.**

| Model risk for COVID-19 | Parameter estimate | Odds ratio | 95% LCI | 95% UCI | p-value |
|---|---|---|---|---|---|
| **Presence of Comorbidities** | | | | | |
| No (Reference) | | | | | |
| Yes | 0.82 | 2.27 | 1.72 | 3.00 | <0.001 |
| **Sex** | | | | | |
| Women (Reference) | | | | | |
| Men | 0.71 | 2.03 | 1.65 | 2.50 | <0.001 |
| **Age** | | | | | |
| <65 of age | | | | | |
| >65 of age | 1.74 | 5.744 | 4.71 | 7.00 | <0.001 |
| Response variable: dead/alive | | | | | |

LCI: lower confidence interval; UCI: upper confidence interval

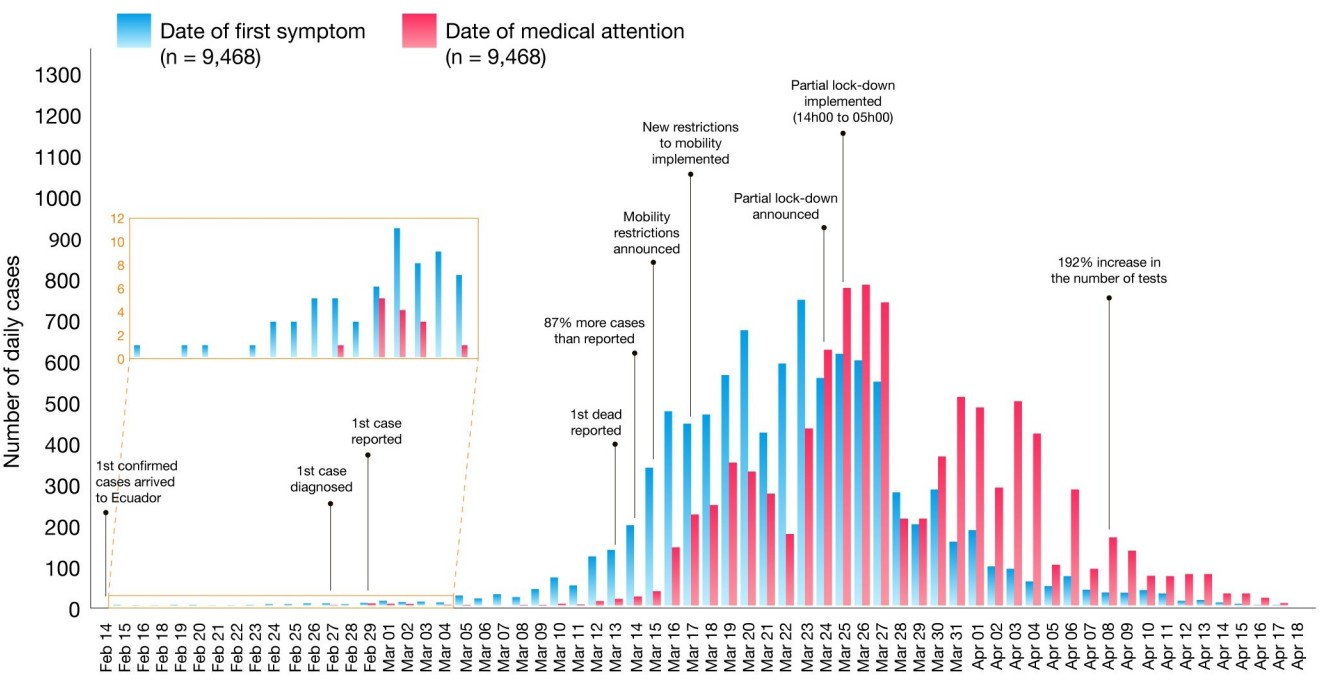

**Fig 3. Epidemiological curves of COVID-19 cases by date of onset of symptoms (blue) and date of diagnosis (red), from 14 February to 18 April 2020.** The first case in Ecuador was diagnosed on 27[th] February but notified on February 29[th].

Guayas, while other provinces, especially those located in the highlands had significantly lower R0 values, including Pichincha with 1.28 (CI: 1.23–1.33), Azuay with 1.36 (CI: 1.24–1.49), and Loja 1.84 (CI: 1.57–2.16).

**Test performed.** In Ecuador, since the start of the outbreak, a total of 19,875 tests have been carried out nationwide, which results in 1,126 tests per million inhabitants. Of the accumulated total, 48% of these were positive and less than 0.5% were not conclusive or suspicious. There were 12,751 unprocessed tests up to April 18[th], 2020 even though nasopharyngeal swab samples were taken (S5 Fig and S1 Table).

The positive testing rate % (PTR%) was 43.19% (95%CI: 37.37–49.01) during the entire period of the study. In March 2020 the PRT% reached a mean of 39.90 (95%CI: 32.83–46.97), meanwhile in April, the PTR% was 44.94% (95%CI: 39.03–50.86).

## Geodemographic distribution

**Result by region and elevation.** We found that coastal regions had higher attack rates than the highlands (p-value: 0.011) and living above 2,500 meters was associated with a lower risk of mortality (RR: 0.63 [CI 95% 0.50–0.79]), compared to populations living at lower altitudes (S6 Fig).

**Daily new cases and deaths by province.** Galapagos (157.3/100,000), Guayas (146.9/100,000), Cañar (49.1/100,000), Santa Elena (37.8/100,000) and El Oro (35.6/100,000) were the provinces with the greatest attack rate per 100,000 (S6 Fig).

In terms of the crude mortality rate, the provinces of El Oro (7.82/100,000), Chimborazo (4.7/100,000), Guayas (4.44/100,000), Santo Domingo de los Tsachilas (4.14/100,000) and Bolivar (2.85/100,000) are the provinces with greatest mortality crude rate per 100,000 (S6 and S7 Figs).

**Burden of disease analysis.** In terms of years of life lost prematurely (YLL), COVID-19 predominantly caused mortality among older adults, especially men. From the start of the outbreak at least 3,207 years were lost prematurely among women and more than 8,847 among men. COVID-19 also caused a loss of 12,112 healthy life years, with an average of 1.27 DALY per case. From the estimated burden of COVID-19, 99.5% is attributable to years of life lost due to premature mortality, with an average of 25.4 YLL per death. The population in Ecuador between 20 to 64 years old contributed to 74.4% of disease burden, followed by the elderly with 24.2% of the burden (S1 Data).

## Discussion

Ecuador has been the worst hit country in the Latin American region from the COVID-19 pandemic [13]. Since the first case confirmed in Ecuador on 27 Feb 2020, at least 9,468 positive COVID-19 cases and 474 deaths were officially registered over a 54-day period.

The images of corpses in the streets and the difficulty of burying the dead occupied the main pages of all the newspapers around the world [13,14]. Overcrowded hospitals and laboratories were overwhelmed by an excessive number of cases, causing a confirmed toll of 474 COVID-19 related deaths and 809 registered suspicious deaths up to April 18[th] 2020 [15]. On April 18, the National Emergency Operations Committee reported 1,061 patients discharged from hospital, 369 hospitalized patients, and 7,564 COVID-19 positives cases, who were stable in-home isolation [16]. Data from the region indicates that Ecuador, along with Brazil, Mexico and Peru are among the top ten most affected countries in Latin America, but when excess mortality is taking into account, Ecuador ranks 1[st] worldwide in the numbers of deaths per million inhabitants [17–19].

At the beginning of the pandemic, Ecuador registered a single case on February 27, when in reality, looking retrospectively, there were already 29 undetected cases. Only two weeks later, only 11 cases were officially registered, but data shows that at least 119 cases were actually circulating within the country (Fig 3). After 54 days, a total of 9,468 positive COVID-19 cases and 474 officially reported deaths, with a sex distribution (55.4% men) that was similar to that reported in China (58% men) and Italy (59.8% men) [20–22].

According to the high PTR% found in Ecuador, the excess mortality that was not correctly diagnosed and reported as COVID-19 and other variables; we inferred that the epidemiological surveillance system and contact tracing strategies were not correctly implemented. At the beginning of the outbreak, every confirmed patient was followed and tracked by the health authority. However, when the number of new COVID-19 cases began to increase exponentially, the system's capacity was overstretched. At the same time, testing capacities were collapsed and only those patients who presented symptoms, those who were in contact with any positive case or those lucky enough to get tested under the surveillance program obtained the diagnosis.

The mortality rate for women (3.35%) is almost half of that for men (6.86%). The median age was 42 in men and 39 in women and the age group most affected was the group from 19–50 years old representing 59.6% of the entire cohort, almost doubling the same age group reported in Italy (24.0%) and very similar to the age distribution from China [21,22]. The reason behind these trends might be due to the fact that Italy has one of the oldest populations in the world. According to the latest data, Italy has 14 million residents over the age of 65 (22%) with an average age of 45.7 years, while in Ecuador the average population is 26.6 years old [7,23,24].

In terms of age, the patients between 0 and 50 years old reported a CFR of 1.6%, compared to 0.4% in Italy, 0.4% in China and 0.6% in Spain from the same age groups [24–26]. When

adjusting for sex, age and the presence of comorbidities, mortality increased significantly among elderly men, which is consistent with other regional studies [21,27].

The existence of comorbidities is linked with augmented age and therefore increased risk of mortality in COVID-19 patients. The age of patients reporting comorbidities in Ecuador was higher than those without comorbidities. In terms of risk, patients with comorbidities had a CFR% of 10.3% in women and 16.9% in men, higher than those without comorbidities and in both sexes CGF% averaged 4%. These findings are equivalent to previous studies that have shown that the presence of comorbidities increases the risk in COVID-19 patients to be admitted to the ICU or die due to this disease [27,28].

In terms of ethnicity, it is important to point out that self-identified Montubios and Indigenous had a CFR% of 14% and 9% respectively, which is surprisingly higher than Mestizos (6%) and other ethnic groups living in Ecuador. This is probably due to reduced healthcare access for vulnerable groups [29]. For influenza, ethnic minorities have the highest estimated fatality rate, most likely due to their social determinants of health, social inequalities and reduced access to health care, especially in rural areas [30,31].

Although the clinical features of COVID-19 are widely studied in moderate and severe hospitalized patients, information on patients with a less severe symptoms is scarce. We collected self-reported data on symptoms from a sample of patients in home isolation. 53% presented with fatigue, 43% headaches and 42% a dry cough which is in agreement with studies from other settings. However a higher proportion of patients reported ageusia (37%) and anosmia (36%), when compared to studies from China (5.1% and 5.6%, respectively) and Italy reporting anosmia in 19.4% of patients [32,33].

In January, alarms went off with the first suspected COVID-19 case in Ecuador, alarms that denoted the poor readiness of the public health system that took more than 15 days to rule-out a highly suspicious COVID-19 patient, who died with the diagnosis of hepatitis B and atypical pneumonia [34,35]. At the beginning of the outbreak, the World Health Organization (WHO) emphasized the importance of testing capabilities in order to improve contact tracing and diseases detection worldwide (S5 Fig). Nevertheless, Ecuador has not enough capabilities to perform molecular diagnosis (RT-PCR), limiting epidemiological surveillance strategies and contact tracing [17,36]. For this reason, the number of samples taken exceeds the local molecular diagnosis capabilities, resulting in testing delays [13,14]. Despite the limitations, in some areas of the country, especially richer areas, such as Samborondon, testing rates are as high as those seen in Iceland or even close to those seen in the US (S1 Table and S1 Data) [37]. This trend might be driven by those patients who were able to pay for their diagnosis and their medical treatment. Therefore, their high attack rate is interpreted as better access to health resources and that is why mortality is also low among these cantons. Although social status and monthly income was not assessed, in Ecuador, blue collar workers have less monetary income than health care and white-collar workers [38,39]. In Ecuador, it is hypothesized that labor workers and the unemployed have less access to health services, therefore the high case fatality rates among these groups is significantly higher than those in more privileged positions such as politicians, medical doctors or health workers (Table 2). These results could indicate that poor working conditions, poverty and therefore limited access to health services could be linked to the high mortality rates reported among the most vulnerable groups, a situation described previously [40]. The working environment and the confinement was also associated with a high prevalence of persons presenting at least mild symptoms of anxiety (58.1%) and depression (52.6%)[41].

Low testing capabilities and high number of suspicious COVID-19 deaths likely distort the calculation of age-specific attack and mortality rates across cantons. We found a median delay of two days (IQR: 1–7) between the day of sampling to the day of notification and a low number of reported tests (test per million people and the overall count) in the region [17].

Therefore, the lack of molecular testing capabilities causes a high positive test rate comparing with other countries in the region [17]. This impacts the ability of contact tracing and other prevention strategies to interrupt SARS-CoV-2 transmission [42]. As evidenced by the high R0 in the provinces closest to the start of the pandemic, where prevention measures were weak and scarce, due to multiple social and political factors. Likewise, other Latin American countries with deficient health policies and lack of timely actions, such as Peru and Brazil, obtained a higher average transmission potential of 2.97[43,44] and 3.1[45,46], respectively.

In terms of elevation we found that people living in cantons and provinces located in the highlands had lower attack rates and lower mortality rates than the populations living at lower altitudes. We suggest that the virus spreads faster at lower altitudes due to demographic and cultural differences. We hypothesized that harsh weather (cold and windy) invites people to stay indoors while at sea level and due to warmer tropical climate, people tend to be more socially active and have greater mobility than the highlands [47]. Additionally, demographic density is lower in the highlands according to the latest INEC data in Ecuador, a situation that might influence the speed of the spread in terms of viral transmission.

Analyzing the years of life lost due to SARS-CoV-2 is challenging due to the uncertainty around the length of the different phases of the disease and the clinical spectrum of the severity of the disease. These uncertainties make it difficult to estimate disability weights that are a crucial component for the DALY calculation. In our study 99.5% of the burden was attributable to the years of life lost due to premature mortality among the study population, with an average of 25.4 YLL per death, and 1.27 DALY per each case of COVID-19. Even though COVID-19 related deaths are higher among elderly populations in developing countries with weakened health systems, mortality is also overstretched among younger populations [48].

The limitations of this study are the lack of specific data during the given time period regarding the type of comorbidities and the presence of clinical manifestations during the initial evaluation. This limitation is also evident when reviewing the follow-up file, which includes only the date of death and the date when the case was discharged and closed. Moreover, the database does not indicate the severity of the disease and whether or not they required hospitalization. Another important fact to consider is that testing was done mainly in symptomatic people diagnosed COVID-19 cases, probably because of limited test availability.

## Conclusions

This is the first study of the epidemiological trends of COVID-19 in Ecuador. The results demonstrate the vulnerability of the health system to contain, mitigate, treat and adequately diagnose this type of new viral disease that spread across the country at a speed that exceeds the speed of response. We found that there were a high number of COVID-19 infections among medical personnel, which likely occurred during the first weeks of the outbreak in the Ecuador. Although the attack rate was high, mortality in this group was very low, which could be due to greater access to healthcare. We also found that the occupations of patients and whether they were unemployed was strongly associated with overall mortality due to COVID-19 in Ecuador, suggesting that poverty is an important driver of the final outcome for this disease. Lastly, strengthening of community-based surveillance and contact tracing response capacities is essential to prevent further advance of the disease within the community.

## Supporting information

**S1 Fig. Timeline of important landmarks during the early phase of the COVID-19 Ecuadorian pandemic in Ecuador.**
(TIF)

**S2 Fig. Reported symptoms by frequency of presentation among mild to moderate patients with a RT-PCR positive COVID-19 test.**
(TIF)

**S3 Fig. Epidemiological distribution of 11 imported and 132 communitarian transmission with direct travel history among COVID-19 confirmed patients in Ecuador.**
(TIF)

**S4 Fig. Confirmed COVID-19 cases and deaths linked to suspected and confirmed COVID-19, during the first 58 days of the outbreak**
(TIF)

**S5 Fig. Testing for COVID-19 in Ecuador.** Number of RT-PCR tests performed since the day of first reporting (February 27th 2020) that were suspected (red), positive (blue), negative (black) and tests not processed (grey).
(TIF)

**S6 Fig. Attack rate and Crude mortality rate by canton in Ecuador during the COVID-19 pandemic (February-April 2020).**
(TIF)

**S7 Fig. Daily number of new confirmed cases and deaths per province due to COVID-19, Ecuador February-April 2020.**
(TIF)

**S1 Table. RT-PCR testing capabilities in Ecuador.** Number of suspected, negative and positive tests performed, and positive testing rates (PTR%).
(DOCX)

**S1 Data. Dataset with further detail**
(XLSX)

## Acknowledgments

The authors thank Rebeca Bravo who was very keen in editing our maps for this publication as well as the staff from the National Direction of Epidemiology for sharing the official database after fulfilling their requirements. We greatly appreciate the work carried out by Dr. Ignacia Paez and her team of Community Mental Health professionals from the Ministry of Public Health, for their role of distributing the self-reporting symptomology tool among the entire cohort of patients who were under constant monitoring from the official epidemiological department of the MoH.

## Author Contributions

**Conceptualization:** Esteban Ortiz-Prado, Clara Paz.

**Data curation:** Esteban Ortiz-Prado, Katherine Simbaña-Rivera, Ana Maria Diaz, Alejandra Barreto, Carla Moyano, Vannesa Arcos, Eduardo Vásconez-González, Fernanda Simbaña-Guaycha, Raúl Fernández-Naranjo, Rachel Lowe.

**Formal analysis:** Esteban Ortiz-Prado, Katherine Simbaña-Rivera, Lenin Gómez Barreno, Martin Molestina-Luzuriaga, Raúl Fernández-Naranjo, Aquiles R. Henriquez-Trujillo, Rachel Lowe.

**Funding acquisition:** Esteban Ortiz-Prado.

**Investigation:** Esteban Ortiz-Prado, Fernanda Simbaña-Guaycha.

**Methodology:** Esteban Ortiz-Prado, Katherine Simbaña-Rivera, Lenin Gómez Barreno, Raúl Fernández-Naranjo.

**Project administration:** Ana Maria Diaz, Alejandra Barreto, Carla Moyano, Vannesa Arcos.

**Software:** Martin Molestina-Luzuriaga, Javier Feijoo, Andrés López-Cortés.

**Supervision:** Esteban Ortiz-Prado, Rachel Lowe.

**Validation:** Esteban Ortiz-Prado, Lenin Gómez Barreno, Aquiles R. Henriquez-Trujillo, Andrés López-Cortés, Isabel Fletcher, Rachel Lowe.

**Visualization:** Esteban Ortiz-Prado, Eduardo Vásconez-González, Martin Molestina-Luzuriaga, Javier Feijoo, Andrés López-Cortés.

**Writing – original draft:** Esteban Ortiz-Prado, Katherine Simbaña-Rivera, Lenin Gómez Barreno, Eduardo Vásconez-González, Rachel Lowe.

**Writing – review & editing:** Esteban Ortiz-Prado, Clara Paz, Aquiles R. Henriquez-Trujillo, Lila Adana, Isabel Fletcher, Rachel Lowe.

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
