## [Decision Letter · Decision Letter 0]

8 Aug 2020

Dear Dr. Ortiz-Prado,

Thank you very much for submitting your manuscript "Epidemiological, socio-demographic and clinical features of the early phase of the COVID-19 epidemic in Ecuador" for consideration at PLOS Neglected Tropical Diseases. As with all papers reviewed by the journal, your manuscript was reviewed by members of the editorial board and by several independent reviewers. In light of the reviews (below this email), we would like to invite the resubmission of a significantly-revised version that takes into account the reviewers' comments. 

We cannot make any decision about publication until we have seen the revised manuscript and your response to the reviewers' comments. Your revised manuscript is also likely to be sent to reviewers for further evaluation.

Sincerely,

Victoria J. Brookes

Deputy Editor

Victoria Brookes

Deputy Editor

Reviewer's Responses to Questions

**Key Review Criteria Required for Acceptance?**

**Methods**

-Are the objectives of the study clearly articulated with a clear testable hypothesis stated?

-Is the study design appropriate to address the stated objectives?

-Is the population clearly described and appropriate for the hypothesis being tested?

-Is the sample size sufficient to ensure adequate power to address the hypothesis being tested?

-Were correct statistical analysis used to support conclusions?

-Are there concerns about ethical or regulatory requirements being met?

Reviewer #1: Yes

Reviewer #2: 1) Describe the epidemiological surveillance system in Ecuador. Where were the cases diagnosed? It seems that most of the cases were severe enough that required medical attention/hospitalization? Were any cases asymptomatic at the time of testing?

2) Could you add information about the testing rates over the course of the epidemic? What was the positivity rate?

**Results**

-Does the analysis presented match the analysis plan?

-Are the results clearly and completely presented?

-Are the figures (Tables, Images) of sufficient quality for clarity?

Reviewer #1: Some issues with data presentation are noted in General Comments

Reviewer #2: 4) Since authors have the epi curve by date of symptoms onset, they could estimate the early reproduction number and assess the impact of interventions on the transmission rate. This addition would make the paper more interesting.

**Conclusions**

-Are the conclusions supported by the data presented?

-Are the limitations of analysis clearly described?

-Do the authors discuss how these data can be helpful to advance our understanding of the topic under study?

-Is public health relevance addressed?

Reviewer #1: Yes

Reviewer #2: (No Response)

**Editorial and Data Presentation Modifications?**

Reviewer #1: Please see General Comments

Reviewer #2: (No Response)

**Summary and General Comments**

Reviewer #1: In their manuscript, Ortiz-Prado et. al. present a cross-sectional analysis of the initial cases of COVID-19 in Ecuador. The data is interesting and provides insight into the trend of COVID-19 early in the transmission in South America, a region that has had a distinct lack of data transparency thus far. Overall, the manuscript tis useful and the data is clear; however, several issues and comments should be addressed and limitations need to be fully considered. 

1) Please consider a full edit for English tense, grammar, and syntax to remain consistent throughout the entire manuscript. 

2) Please remove statements of primacy. 

3) Introduction: As the pandemic is a very dynamic situation, please qualify COVID-19 numbers with a respective date (and update to most recent data available) in order to maintain accurate reporting in publication over the current global crisis. 

4) Introduction: airborne transmission and it’s association with SARS-Co-2 spread is still under debate (though data does indicate it is possible). Please also consider using the term “droplet transmission.”

5) Please indicate how individuals were diagnosed and confirmed with SARS-CoV-2 infection. 

6) Figure 1: Only 143 individuals are reported in this figure? How can community-transmitted cases have a geographical origin other than Ecuador? Overall, this figure is a bit confusing and is not discussed in the text; therefore, it may be better to discuss or to leave out entirely. 

7) Figure 2: blue and purple colors do not translate to grayscale, please consider revising

8) Based on the limited power, (n=84) for vaccinated individuals, it may not be advisable to indicate that mortality risk was lower in this population, especially since statistical significance was not achieved. 

9) Figure 3 is marked as Figure 1 in the legend. Please correct. 

10) As the pandemic continues, and Ecuador is not experiencing >70,000 cases of COVID-19 wirh no indication of curve reduction: it may be better to refer to all data “during the given time period” or with the specific dates as to ensure accurate usage of the facts and figures calculated. 

11) Is the comparison between the coastal and highland regions due to any issues of bias in the reported data or how the virus was introduced into the country and spread? 

12) Discussion: may want to modify the statement about Ecuador and Panama affectedness with the current situation in Brazil, etc…again, it is understandable that these rankings and figures are extremely difficult to keep up with in terms on an ongoing global pandemic.

13) While the lack of data on comorbidities and presence of clinical manifestations does represent a significant limitation, there are a number of other equally important limitations set out in the discussion (ie: information on testing and access to healthcare) and others which are not discussed (ie: unknown climate, viral, or social factors) which are also equally important. It may be better to restructure the discussion to address all possible limitations in a highly succinct way. 

14) Please put consider including statistical results in Table 1, thus mitigating the need for Table 2 and making data comparison easier to analyze.

Reviewer #2: In this paper authors present the results of an interesting descriptive analysis of the epidemiological characteristics of a series of confirmed cases with COVID-19 in Ecuador. Overall the paper is well written and the findings reported here add to our understanding of the impact of the pandemic in Latin America and beyond. However, I have a few comments that need to be addressed in a revised version:

1) Describe the epidemiological surveillance system in Ecuador. Where were the cases diagnosed? It seems that most of the cases were severe enough that required medical attention/hospitalization? Were any cases asymptomatic at the time of testing?

2) Could you add information about the testing rates over the course of the epidemic? What was the positivity rate?

3) Authors need to put their findings in context. There are now a few studies reporting key transmission and epidemiological characteristics (including the adjusted CFR and the reproduction number) of the pandemic impact in Latin American countries including Brazil, Peru, and Chile.

4) Since authors have the epi curve by date of symptoms onset, they could estimate the early reproduction number and assess the impact of interventions on the transmission rate. This addition would make the paper more interesting.

5) Authors should update the analysis using the most recent data available.

PLOS authors have the option to publish the peer review history of their article (what does this mean?). If published, this will include your full peer review and any attached files.

Reviewer #1: No

Reviewer #2: No
---

## [Editor Report · Decision Letter 1]

16 Oct 2020

Dear Dr. Ortiz-Prado,

Thank you very much for submitting your manuscript "Epidemiological, socio-demographic and clinical features of the early phase of the COVID-19 epidemic in Ecuador" for consideration at PLOS Neglected Tropical Diseases. As with all papers reviewed by the journal, your manuscript was reviewed by members of the editorial board and by several independent reviewers. The reviewers appreciated the attention to an important topic. Based on the reviews, we are likely to accept this manuscript for publication, providing that you modify the manuscript according to the review recommendations. 

Please consider the additional changes to the manuscript listed below.

Sincerely,

Victoria Brookes

Deputy Editor

We thank the authors for their responses. Reviewers have requested the following additional changes to the manuscript:

Please incorporate the complete methodology and additional results into the methods and results in the main paper.

Line 67 ‘covid19’ to ‘COVID-19

Line 105 Please re-write this sentence because the meaning is not clear:

‘We computed the time distribution from hospital admission to the time of using the methodology reported by Russel 2020.’

Line 131: Please check the figure numbers throughout the manuscript.

Line 139: 

‘These differences were compared with the national distribution for each ethnic group according to the 2010 national census.’

Where are the results for these comparisons? If this is included in Table 1, please make this clear (here, and in the Table labels).

Line 146: ‘Had’ to ‘have’

Line 157: The meaning of this sentence is not clear. Please re-write and check the figure number (for previous figures too – Figures 1 and 2 appear to be labelled the wrong way round).

‘When 158 observed retrospectively using the date of the onset of symptoms, 29 undocumented patients were already sick but only 6 were diagnosed, trend that continued to be high until March 24th 159 , 160 the day when the lock-down was implemented in Ecuador (Figure 2).’

Line 211:

‘At the beginning of the pandemic, Ecuador registered a single case on February 27, when in 212 reality, looking retrospectively, there were already 19 undetected cases. Only two weeks later, 213 there were 11 officially registered cases but at least 119 undetected cases (Figure 3).’

Where is Figure 3, and please explain more clearly how numbers of undetected cases were known (if numbers are known, was this retrospective case ascertainment?). If these are the results of this analysis, move this sentence and explanation to the results. 

Line 217:

Please clarify this sentence (‘was denoted by’?) and state how this is known (is it inferred from the findings of this study, or other evidence?): ‘The epidemiological surveillance system and contact tracing strategy denoted a lack of planning and the lack of enough personnel and resources.’

Line 218 ‘begging’ to ‘beginning’

Line 279: ‘calculation age-specific attack’ to ‘calculation of age-specific attack’

Line 295: ‘a situation’
---

## [Editor Report · Decision Letter 2]

27 Oct 2020

Dear Dr. Ortiz-Prado,

Thank you very much for submitting your manuscript "Epidemiological, socio-demographic and clinical features of the early phase of the COVID-19 epidemic in Ecuador" for consideration at PLOS Neglected Tropical Diseases. As with all papers reviewed by the journal, your manuscript was reviewed by members of the editorial board and by several independent reviewers. The reviewers appreciated the attention to an important topic. Based on the reviews, we are likely to accept this manuscript for publication, providing that you modify the manuscript according to the review recommendations. 

There are still some remaining unusual items in the supplementary material. Finally please update as follows: 

Additional Conclusions - to manuscript

Acknowledgement - remove

Authors' Contributions - can leave

Disclosure Statement - remove

Funding Sources - remove

Sincerely,

Victoria J. Brookes

Deputy Editor

Victoria Brookes

Deputy Editor

Finally, from the supplementary material, please update as follows: 

Additional Conclusions - to manuscript

Acknowledgement - remove

Authors' Contributions - can leave

Disclosure Statement - remove

Funding Sources - remove
---

## [Editor Report · Decision Letter 3]

5 Nov 2020

Dear Dr. Ortiz-Prado,

We are pleased to inform you that your manuscript 'Epidemiological, socio-demographic and clinical features of the early phase of the COVID-19 epidemic in Ecuador' has been provisionally accepted for publication in PLOS Neglected Tropical Diseases.

Best regards,

Victoria J. Brookes

Deputy Editor

Victoria Brookes

Deputy Editor

---

## [Editor Report · Acceptance letter]

17 Dec 2020

Dear Dr. Ortiz-Prado,

We are delighted to inform you that your manuscript, "Epidemiological, socio-demographic and clinical features of the early phase of the COVID-19 epidemic in Ecuador," has been formally accepted for publication in PLOS Neglected Tropical Diseases.

Best regards,

Shaden Kamhawi

co-Editor-in-Chief

Paul Brindley

co-Editor-in-Chief
